# Gymnasium:
# A Standard Interface for Reinforcement Learning Environments

## Abstract

Reinforcement Learning (RL) is a continuously growing field that has the potential to revolutionize many areas of artificial intelligence. However, despite its promise, RL research is often hindered by the lack of standardization in environment and algorithm implementations. This makes it difficult for researchers to compare and build upon each other's work, slowing down progress in the field. Gymnasium is an open-source library that provides a standard API for RL environments, aiming to tackle this issue. Gymnasium's main feature is a set of abstractions that allow for wide interoperability between environments and training algorithms, making it easier for researchers to develop and test RL algorithms. In addition, Gymnasium provides a collection of easy-to-use environments, tools for easily customizing environments, and tools to ensure the reproducibility and robustness of RL research. Through this unified framework, Gymnasium significantly streamlines the process of developing and testing RL algorithms, enabling researchers to focus more on innovation and less on implementation details. By providing a standardized platform for RL research, Gymnasium helps to drive forward the field of reinforcement learning and unlock its full potential.

## 1 Introduction

With the development of Deep Q-Networks (DQN) (Mnih et al., 2013), the field of Deep Reinforcement Learning (DRL) has gained significant popularity as a promising paradigm for developing autonomous AI agents. Throughout the last decade, DRL-based approaches managed to achieve or exceed human performance in many popular games, such as Go (Silver et al., 2017), DoTA 2 (Berner et al., 2019) or Starcraft 2 (Vinyals et al., 2019). During this time, OpenAI Gym (Brockman et al., 2016) emerged as the de facto standard open source API for DRL researchers. Its simple structure and quality of life features made it possible to easily implement a custom environment that is compatible with existing algorithm implementations. Gymnasium is the updated and maintained version of OpenAI Gym. In this paper, we outline the main features of the library, the theoretical and practical considerations for its design, as well as our plans for future work. We hope that this work removes barriers from DRL research and accelerates the development of safe, socially beneficial artificial intelligence.

## 2 Related Work

Due to its influential positioning, Gymnasium enjoys a wide ecosystem of compatible libraries. In this section, we describe some of the most prominent RL libraries that can be used with Gymnasium, as well as alternative API libraries.

---

[†]Co-first authors

[‡]Contribution occurred prior to joining Meta

[*]Alphabetically ordered

## 2.1 TRAINING LIBRARIES

**Stable Baselines3 (SB3)** (Raffin et al., 2021) is a popular library providing a collection of state-of-the-art RL algorithms implemented in PyTorch. It builds upon the functionality of OpenAI Baselines (Dhariwal et al., 2017), aiming to deliver reliable and scalable implementations of algorithms like PPO, DQN, and SAC.

**CleanRL** (Huang et al., 2022) is designed to provide clean, minimalistic implementations of RL algorithms. It focuses on simplicity and transparency, making it easier for researchers to understand and experiment with different RL techniques.

**Tianshou** (Weng et al., 2021) is a versatile library for RL research that supports various training paradigms, including off-policy, on-policy, and multi-agent settings. It offers a modular design that allows users to easily customize and extend the library's components.

**Ray Rllib** (Liang et al., 2018) is part of the Ray ecosystem and is known for its scalability and support for distributed RL training. Rllib provides a diverse range of algorithms and tools for both single-agent and multi-agent scenarios.

**Dopamine** (Castro et al., 2018) is a research framework developed by Google for experimenting with reinforcement learning algorithms. It is designed to provide a clean, minimalistic codebase that focuses on implementing and evaluating RL algorithms such as DQN and its variants. Dopamine emphasizes reproducibility and simplicity, offering well-structured, modular components that make it easy for researchers to implement and test new algorithms.

**d3rlpy** (Seno & Imai, 2022) is a user-friendly offline reinforcement learning library built on PyTorch. It emphasizes ease of use while providing robust implementations of offline RL algorithms such as CQL, BCQ, and CRR. d3rlpy supports both continuous and discrete action spaces and offers tools for benchmarking offline datasets, making it a valuable resource for researchers working on offline RL or real-world applications where interaction with the environment is limited.

**TorchRL** (Bou et al., 2023) is an RL library developed under the PyTorch ecosystem, aiming to provide modular, flexible, and efficient tools for RL research. It includes a wide range of utilities for building and training RL agents, such as prebuilt environments, efficient data collection pipelines, and distributed training support.

## 2.2 ALTERNATIVE API LIBRARIES

Several alternative libraries offer different approaches to defining and interacting with RL environments, providing valuable insights and capabilities beyond Gymnasium.

**Dm_env** (Muldal et al., 2019) is part of the DeepMind Control Suite and offers a lightweight API for RL environments. It emphasizes a clean, functional design and is used for evaluating algorithms in a controlled manner. While similar in some aspects to Gymnasium, dm_env focuses on providing a minimalistic API with a strong emphasis on performance and simplicity.

**PettingZoo** (Terry et al., 2021) is designed for multi-agent RL environments, offering a suite of environments where multiple agents can interact simultaneously. It builds on concepts from Gymnasium but extends its capabilities to support complex multi-agent scenarios, making it an important tool for research in cooperative and competitive settings.

**OpenAI Gym** (Brockman et al., 2016), the predecessor to Gymnasium, remains a widely used library in RL research. Gymnasium is built upon and extends the Gym API, retaining its core principles while introducing improvements and new features. Gym's well-established framework continues to serve as a foundation for many RL environments and algorithms, reflecting its influence on the development of Gymnasium. Gym environments can be automatically converted to Gymnasium environments using Shimmy (Tai et al., 2023).

## 2.3 OTHER TOOLING

**Minari** (Younis et al., 2024) defines a standardized format for offline RL datasets and provides a suite of tools for data management. These tools facilitate collection, ingestion, processing, and

distribution of datasets. Additionally, Minari integrates with a cloud-based repository that hosts a variety of benchmark datasets, enhancing accessibility and reproducibility in offline RL research.

**Metaworld** (Yu et al., 2019) is a benchmark for meta-RL and multi-task learning. It contains 50 robotic manipulation tasks and allows for easy benchmarking of algorithms compatible with the Gymnasium interface.

**Gymnasium Robotics** (de Lazcano et al., 2023) is a collection of robotics environments, including maze path-finding and robot arm manipulation. It provides a multi-goal API compatible with Gymnasium, enabling support for algorithms like Hindsight Experience Replay (Andrychowicz et al., 2018).

**Atari Learning Environment** (Bellemare et al., 2013) is a collection of environments based on classic Atari games. It uses an emulator of Atari 2600 to ensure full fidelity, and serves as a challenging and diverse testbed for RL algorithms.

## 3 BASIC LIBRARY STRUCTURE

```python
import gym

env = gym.make("CartPole-v1")

obs = env.reset()
env.seed(0)

while True:
    action = env.action_space.sample()
    obs, reward, done, info = env.step(action)
    env.render("human")

    if done:
        break
```

(a) OpenAI Gym

```python
import gymnasium as gym

env = gym.make("CartPole-v1", render_mode="human")

obs, info = env.reset(seed=0, options={"low": -0.1, "high": 0.1})

while True:
    action = env.action_space.sample()
    obs, reward, terminated, truncated, info = env.step(action)
    env.render()

    if terminated or truncated:
        break
```

(b) Gymnasium

Figure 1: Comparison of the basic loop in OpenAI Gym and Gymnasium.

At its core, Gymnasium is a collection of interfaces tailored for usage in RL research, so that they can be reused by researchers and developers across various approaches and algorithms. In this section, we briefly describe the main abstractions and interfaces included in Gymnasium.

**Environment** The central abstraction in Gymnasium is the `Env`. It represents an instantiation of an RL environment, and allows programmatic interactions with it. An `Env` roughly corresponds to a Partially Observable Markov Decision Process (POMDP) (Kaelbling et al., 1998), with some

notable differences discussed in Section 4.1. We show the general usage of an `Env` in Figure 1, with the equivalent code in the OpenAI Gym version for comparison.

An `Env` is primarily defined by the following components:

1. `reset` method
2. `step` method
3. Observation space
4. Action space

The `reset` method sets the environment in an initial state, beginning an episode. The `step` method executes a selected action in the environment, and is the only mechanism of moving the simulation forward. The observation space defines the structure of observations that the environment emits, and the action space defines valid actions that can be used in `step`. For a detailed description of the `Env` class, we refer the reader to the documentation.

**Vector Environment**  A secondary, but arguably as important abstraction, is the `VectorEnv`. It represents a batch of identical, but independently running RL environments. In terms of POMDPs, they are identical POMDPs, but with independently sampled initial states, and executing potentially different actions. A `VectorEnv` has a structure analogous to that of an `Env`, with the key difference being that everything is batched across multiple independent environments. This is crucial for modern RL research, as it enables significant performance gains by parallelizing the environment execution, and the policy evaluation.

**Space**  `Space` is an abstraction shared between observation and action spaces. It roughly corresponds to a set, with some structure imposed on it to improve its utility for representing (observation and action) sets relevant to RL environments. We further describe Gymnasium spaces in Section 4.3.

**Registry**  Gymnasium provides a simple way to manage environments via a registry. Any environment can be registered, and then identified via a namespace, name, and a version number. The standard Gymnasium convention is that any changes to the environment that modify its behavior, should also result in incrementing the version number, ensuring reproducibility and reliability of RL research.

## 4 NOVEL FEATURES

Since its creation as a fork of Gym, we extended Gymnasium to include various new tools and features to simplify RL research and make it more reliable. In this section, we describe the main new additions.

### 4.1 FUNCTIONAL API AND POMDPS

While the object-oriented `Env` class is the standard way of using Gymnasium, the library also provides `FuncEnv`, a secondary abstraction with a more functional approach. Beyond just enabling a different programming paradigm, this abstraction has two main advantages: it is more closely connected to the theoretical POMDP formalism, and it enables easy hardware acceleration of implemented environments by using libraries like Jax (Bradbury et al., 2018).

Similarly to `Env`, a `FuncEnv` is defined with several functions:

1. `initial` function: Generates the initial state of the environment.
2. `observation` function: Returns the observation for a given state in the environment.
3. `transition` function: Computes the next state of the environment based on an action.
4. `reward` function: Computes the reward for transitioning from one state to another given an action.
5. `terminal` function: Determines if a state is terminal, indicating the end of an episode.

This is very close to the components of a POMDP. As a reminder, a POMDP is typically defined as a tuple $\mathcal{M} = (\mathcal{S}, \mathcal{A}, T, R, \Omega, O, \mu)$, where

- $\mathcal{S}$ is a set of states of the environment.
- $\mathcal{A}$ is a set of actions available to the agent.
- $T \colon \mathcal{S} \times \mathcal{A} \to \Delta\mathcal{S}$ is the environment transition function, representing its dynamics.
- $R \colon \mathcal{S} \times \mathcal{A} \times \mathcal{S} \to \mathbb{R}$ is the reward function which is used to define the agent's task.
- $\Omega$ is a set of possible observations
- $O \colon S \to \Delta\Omega$ is the observation function mapping states to observations.
- $\mu \in \Delta\mathcal{S}$ is the initial state distribution.

In the usual object-oriented API, we cannot create a one-to-one connection between the `Env` members and methods, and the POMDP components. Instead, the API combines some of the components:

1. `reset` is equivalent to sampling $s \sim \mu$, and returning a sample from $O(s)$
2. `step` takes an action $a \in \mathcal{A}$, computes the next state as $s' \sim T(s, a)$, and then returns the new observation $o \sim O(s')$ and the reward $r = R(s, a, s')$
3. The observation space is equivalent to $\Omega$
4. The action space is equivalent to $\mathcal{A}$

In the case of the `FuncEnv`, the connection is much closer:

1. `initial` function corresponds directly to sampling the initial state from $\mu$, i.e., $s \sim \mu$.
2. `observation` function corresponds to the observation function $O(s)$, returning an observation $o \sim O(s)$ for a given state $s$.
3. `transition` function directly implements the transition function $T(s, a)$, which computes the next state $s' \sim T(s, a)$ given a state $s$ and action $a$.
4. `reward` function implements the reward function $R(s, a, s')$, which calculates the reward based on a given state $s$, action $a$, and the resulting next state $s'$.

The one missing function, `terminal`, does not correspond to any element of the classical POMDP description, but is necessary for practical reasons. We expand on this in the following section.

## 4.2 TERMINATION AND TRUNCATION

In RL theory, it is common to assume that actions can be executed in an environment indefinitely, without stopping. In practical work, this tends to be unrealistic, as researchers only have finite time to perform their experiments. To account for that, Gymnasium introduces the notions of episode **termination** and **truncation**. In OpenAI Gym, these two concepts were not clearly separated, so it is worth expanding on their utility and the distinction between them.

**Termination** is a signal that depends on the state reached after an environment transition. Typically, this represents either success or failure of the task that the RL agent is trying to achieve, but generally, it is any state-dependent reason to end an episode. After reaching a terminal state, `step` cannot be called again until the environment is reset. It is possible for an environment to not have any terminal states, as is common in theoretical RL problems. In the POMDP formalism, the typical way to represent episode termination is entering an absorbing state with 0 reward.

**Truncation** is similar to termination in the fact that it indicates the end of an episode, but instead of being state-based, it is time-based. It is roughly the equivalent of saying "The episode could keep going, but we ran out of time, so we decided to stop it here". In Gymnasium, we provide the `TimeLimit` wrapper that is applied to environments by default, through which developers and users can define a number of steps after which the environment is guaranteed to finish via truncation.

While the difference between termination and truncation may seem minor, it has nontrivial implications for algorithm implementations. For example, in algorithms like REINFORCE (Sutton et al.,

1999) and Proximal Policy Optimization (PPO) (Schulman et al., 2017), a key component is estimating the value of a given state, that is the expected discounted sum of rewards in the rest of the episode. Similarly, in value-based algorithms like Deep Q-Networks (DQN) (Mnih et al., 2015) or Soft Actor-Critic (Haarnoja et al., 2018), the central concept is that of a state-action value, which also depends on future rewards.

If an episode terminates, this means that there are no further rewards to be obtained. This means that in the value estimation process, the value after a terminal state is equal to 0. In contrast, if an episode is truncated, that means that the agent encountered a rather arbitrary cutoff. If not for the truncation, it could have kept acting and accumulating rewards, so in the value estimation algorithm, we use an estimate of the final state's value.

For a practical example, consider a racing environment where the agent has to complete $n$ laps to finish the race as quickly as possible. If the final signal is that of termination, this means that the agent should optimize for that exact number of laps. If its trajectory at the end of the last lap means that it would crash shortly after passing the finish line, that is not a problem – that will never happen. Conversely, if the final signal is truncation, then the agent will be incentivized to act as if it would keep racing. This would result in decreasing the reward obtained in the original timeframe, but acting in a safer, more sustainable manner towards the end of the race.

## 4.3 ALGEBRAIC SPACES

In Gymnasium, observation and action spaces can be broadly split into two categories – **fundamental** and **composite**. As the name suggests, composite spaces are composed of one or more subspaces, whereas fundamental spaces cannot be divided this way. There are the following fundamental spaces:

1. Box – for multidimensional arrays typically containing real numbers
2. Discrete – for single integers
3. MultiBinary – for sequences of binary values
4. MultiDiscrete – for sequences of integers
5. Text – for strings

There are also the following composite spaces:

1. Dict – for (Python) dictionaries of spaces
2. Tuple – for tuples of spaces
3. Sequence – for variable-length sequences of spaces
4. Graph – for graphs of spaces (both nodes and edges)
5. OneOf – for disjoint unions of spaces

Considering just the fundamental spaces, and the Tuple[*] and OneOf composite spaces, we observe that Gymnasium spaces mirror the structure of Algebraic Data Types. We can take any collection of spaces and combine them into a Tuple to obtain a product type – an element of a Tuple space must contain an element from each of its subspaces. We can also combine them into a OneOf space, equivalent to a sum type, which contains an element from one of its subspaces, along with the index of that subspace.

## 4.4 VECTORIZATION

Gymnasium puts an emphasis on treating vectorized environments as first class citizens, on par with individual environments. This is because vectorization is a common optimization technique in RL research, enabling significant performance gains without major changes to the algorithm implementation.

---

[*]Or, equivalently, Dict. The difference between the two is the customizability of dictionary keys for the sake of usability.

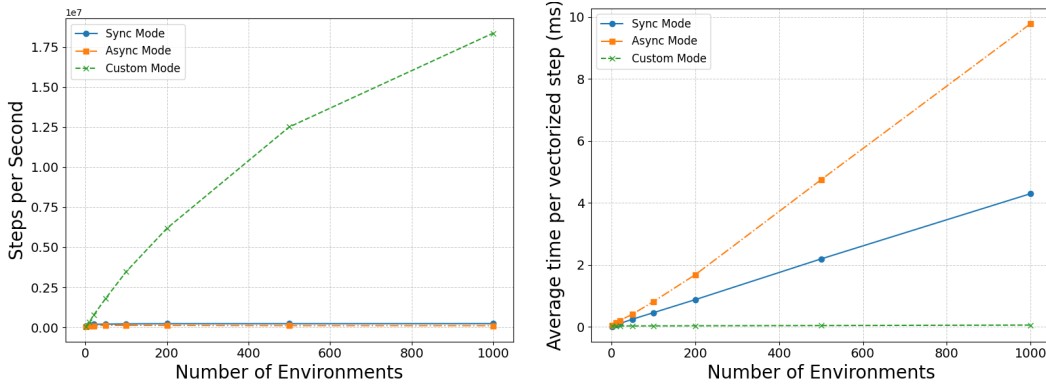

(a) Individual environment steps per second. Higher is better.

(b) Average time to perform a single batched step. Lower is better.

Figure 2: Performance comparison of different vectorization modes on the simple Cartpole environment, as a function of the number of vectorized environments. The overhead of spawning subprocesses leads to poor performance of the `Async` vectorization as compared to the naive `Sync` vectorization. Custom, NumPy-based vectorization outperforms both environment-agnostic methods.

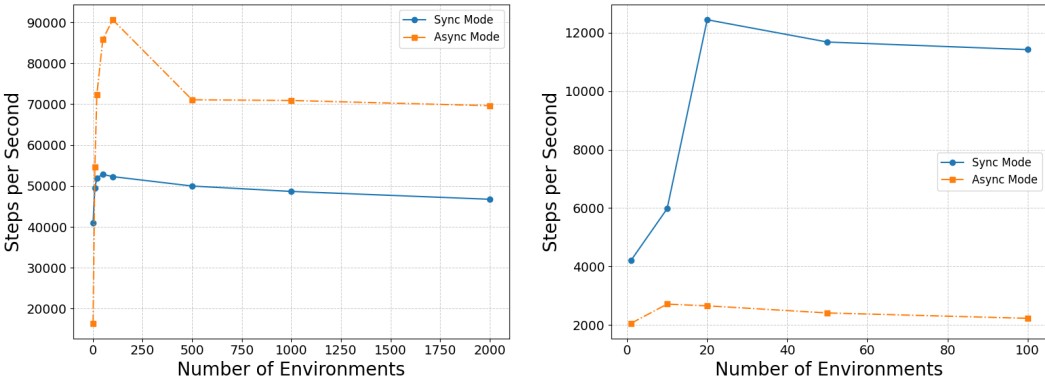

(a) Individual environment steps per second, executed on a MacBook Pro. Higher is better.

(b) Individual environment steps per second, executed on Google Colab. Higher is better.

Figure 3: Performance comparison of different vectorization modes on the more complex Lunar Lander environment. On a system with high RAM (M3 MacBook Pro with 128 GB of RAM), `Async` vectorization provides performance benefits by running environments in parallel. Conversely, on weaker hardware (Google Colab), the subprocess overhead remains significant and significantly degrades `Async` performance, leading to `Sync` vectorization being the better option.

By default, Gymnasium supports two vectorization modes, as well as a custom mode that can be defined for each environment separately. `SyncVectorEnv` vectorizes arbitrary environments in the simplest way – it runs them in sequence and batches the results. `AsyncVectorEnv` is the second method of vectorizing arbitrary environments, by running each of them in its own subprocess.

The relative performance of `Sync` and `Async` vectorization depends on many factors, notably the complexity of the environment itself and the properties of the hardware that it is run on. For simple and computationally inexpensive environments like Cartpole, the overhead introduced by creating subprocesses in `Async` vectorization exceeds the computational needs of the environment. This means that `Sync` significantly outperforms it, and we can obtain even better performance by implementing a vectorized environment using NumPy vectorization. We show this in Figure 2.

With more complex environments like the Lunar Lander, the picture gets more complicated. In this case, on a machine with enough memory, subprocess-based `Async` vectorization can provide

a significant performance boost, as spawning new processes is not that impactful. On a weaker machine, the subprocess overhead remains impactful, resulting in `Sync` vectorization yielding better performance. We show this in Figure 3.

In conclusion, the choice of vectorization is far from a trivial one, and the `VectorEnv` abstraction in Gymnasium enables easy experimentation between the two built-in methods, as well as custom user-defined approaches. This allows every user to tailor their approach to obtain the best performance for their use case, without having to change the rest of their code.

## 5 BUILT-IN ENVIRONMENTS

Gymnasium includes a collection of well-tested environments that are fully compliant with the API. Those can serve as references for new environment implementations, or as simple testing grounds for algorithm development. Here, we describe these environments.

**Toy text** There are four toy text environments: Blackjack, Taxi, Cliff Walking and Frozen Lake. These are discrete environments that can be modeled as tabular MDPs, and as such can serve as testbeds for RL algorithms that do not rely on neural network for state representation.

**Classic control** In the classic control suite, there are four environments: Cartpole, Acrobot, Mountain Car, and Pendulum. Each of them has a fairly simple physics simulation at its core, a continuous observation space, and either a discrete or continuous action space. These environments tend to serve well as quick and simple evaluations for new algorithm implementations. A notable exception is Mountain Car, which is often difficult to solve without a deliberate exploration mechanism.

**Box2D** There are three environments using the open-source Box2D physics engine: Bipedal Walker, Car Racing, and Lunar Lander. These environments use a more complex physics simulation, including components like collision detection and contact forces (both of which are neglected in the classic control environments). These represent another step up in complexity, posing a non-negligible, but still very tractable challenge for RL algorithm implementations.

**Mujoco** MuJoCo (Todorov et al., 2012) environments are a set of eleven physics-based continuous robot control environments. These environments simulate more complex physical interactions, including multi-body dynamics and contact forces, offering a more realistic and high-dimensional setting compared to the previously described environments. Each of these environments involves controlling a simulated agent to achieve tasks such as locomotion or balance in a continuous state and action space. The MuJoCo environments are often used as benchmarks for deep reinforcement learning algorithms due to their higher dimensionality, non-linear dynamics, and need for precise control.

## 6 SUMMARY

This paper introduces Gymnasium, an open-source library offering a standardized API for RL environments. Building on OpenAI Gym, Gymnasium enhances interoperability between environments and algorithms, providing tools for customization, reproducibility, and robustness. It is compatible with a wide range of RL libraries and introduces various new features to accelerate RL research, such as an emphasis on vectorized environments, and an explicit interface for functional environments that can be hardware-accelerated. Gymnasium includes a suite of benchmark environments ranging from finite MDPs to MuJoCo simulations, streamlining RL algorithm development and evaluation.

### 6.1 FUTURE WORK

Going forward, our goal is to promote the usage of Gymnasium among RL researchers as the "glue" between otherwise separate projects. We will continue improving the provided tooling, simplifying researchers' work and improving their efficiency. By integrating future theoretical and experimental

findings, we will enable their usage with simple code changes, compatible with the rest of Gymnasium code. In the long term, we hope to foster a community around a stable and comprehensive ecosystem of RL tooling, with the goal of accelerating advancements in safe and beneficial AI research.

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
