# OpenReview forum: "Gymnasium: A Standard Interface for Reinforcement Learning Environments"
_ICLR.cc/2025/Conference — Submitted to ICLR 2025_

### Official Review · Reviewer_uUKD · 2024-10-16

**Soundness:** 4
**Presentation:** 2
**Contribution:** 4
**Rating:** 8
**Confidence:** 5

**Summary:**

This paper introduces Gymnasium, an open-source library that provides a standard API for building RL environments. As a maintained fork of OpenAI Gym, Gymnasium offers enhanced features and interoperability across RL environments and training algorithms. Gymnasium has significantly outperformed the Gym in terms of built-in environments, application ecosystem, and operating efficiency. There is no doubt that Gymnasium has become the new standard interface in the RL community, which is expected to continue to drive forward the field.

**Strengths:**

It is natural to compare Gymnasium and the OpenAI Gym to highlight its strengths:

- **Active developer community**. As a maintained fork, Gymnasium has gained 7.1k stars, 450+ issues and, 750+ PRs on GitHub, which demonstrates a really active community. This open-source collaboration will make Gymnasium a comprehensive library that aligns with the latest direction of RL.

- **New `.step()` logic**. Gymnasium introduces the notions of episode
  **termination** and **truncation**. **Termination** ends an episode based on the agent reaching a specific state, like winning or failing, signaling the task is complete. **Truncation** stops an episode after a set time or a number of steps, even if the task isn't finished, helping to limit long-running tasks for practical experiments. Both concepts are important for controlling episode length and ensuring efficient learning in RL.

- **More built-in environments**. As compared to Gym, Gymnasium provides more built-in environments that cover more research areas, such as offline RL and meta RL. Gymnasium's team also integrated many old and important environments, such as MiniGrid, Procgen, and Meta-World.

- **Performance optimization**. The library's emphasis on **vectorized environments** allows researchers to parallelize experiments and improve performance without major changes to algorithms. The performance comparisons presented in the paper show significant gains with proper vectorization modes.

- **Functional API**. The introduction of a *FuncEnv* offers flexibility in programming paradigms and is better aligned with the POMDPs. It also allows for efficient usage of hardware accelerators like JAX.

**Weaknesses:**

Please refer to the questions part.

**Questions:**

Based on the presented paper and the reported progress on GitHub, I have the following questions:

- Indeed, Gymnasium is a great successor to Gym. But it is undeniable that there are still a large number of projects using Gym due to various reasons like old environments or dependencies. How will Gymnasium promote further popularization in the community?

- On the other hand, how does Gymnasium achieve better backward compatibility with old code? A section about the code compatibility should be added.

- In Gymnasium, the `.reset()` is re-implemented, and the `.seed()` method is deprecated. I don't see the necessity for this change, could you elaborate it further?

- In Section 2.1, more new libraries should be included, such as d2rlpy [1] (for offline RL), TorchRL [2], and RLLTE [3]. Dopamine is a bit too old.

- I did't see the future work section, which is quite important for the paper.

- Code examples should be included, which also need to demonstrate the difference between Gym and Gymnasium.

- I strongly recommend the authors to make a chart to illustrate the structure and new logic of Gymnasium.

- For Section 5, using a table that illustrates the built-in envs and related official envs (e. g., MiniGrid and Procgen) would be better. The authors can also highlight the potential research targets of these envs, such as Procgen is procedurally-generated and MiniGrid is for exploration.

- The performance benchmarks focus on relatively simple environments such as **Cartpole** and **Lunar Lander**. It would better to include a broader range of environments, particularly more challenging and high-dimensional ones, to demonstrate Gymnasium's true potential.


I'm pleased to increase my rating if my questions are addressed.

References

[1] Seno T, Imai M. d3rlpy: An offline deep reinforcement learning library[J]. Journal of Machine Learning Research, 2022, 23(315): 1-20.

[2] Bou A, Bettini M, Dittert S, et al. TorchRL: A data-driven decision-making library for PyTorch[C]//The Twelfth International Conference on Learning Representations.

[3] Yuan M, Zhang Z, Xu Y, et al. RLLTE: Long-Term Evolution Project of Reinforcement Learning[J]. arXiv preprint arXiv:2309.16382, 2023.

---

> ### Author Response · Authors · 2024-11-21
>
> We would like to thank the reviewer for their valuable comments, and for their very kind words about our work.
>
> “How will Gymnasium promote further popularization in the community?”
>
> To some extent, inertia is inevitable - it’s hard to expect an old and sometimes-used project to undergo a rewrite to Gymnasium. That being said, as more and more libraries become compatible with Gymnasium, we expect users to migrate to it over time, so that they can keep using up-to-date tooling.
>
> “how does Gymnasium achieve better backward compatibility with old code?”
>
> In previous versions, we had explicit features enabling compatibility with older versions, but we removed it to maintain the cohesion of the API. Instead, we provide an additional library called Shimmy which allows wrapping an existing Gym environment into a Gymnasium-compatible object. We added a mention of this in the paper.
>
>
> “In Gymnasium, the .reset() is re-implemented, and the .seed() method is deprecated. I don't see the necessity for this change, could you elaborate it further?”
>
> This was a fairly difficult decision, but it was done to keep the API as consistent as possible. While everything is simple in python-native environments that use Gymnasium’s built-in RNG utilities, it can be hard to specify what “seeding” an environment in the middle of an episode should even mean, and in the case of some environments (e.g. Atari), it was impossible. For this reason, we considered a constraint that .seed() can only be called immediately after environment reset, and from this, we decided to just merge the two methods.
>
>
>
> “In Section 2.1, more new libraries should be included, such as d2rlpy [1] (for offline RL), TorchRL [2], and RLLTE [3]. Dopamine is a bit too old.”
>
> “I did't see the future work section, which is quite important for the paper.”
>
> “Code examples should be included, which also need to demonstrate the difference between Gym and Gymnasium.”
>
> Thank you, we updated the paper to address this.
>
>
>
> “The performance benchmarks focus on relatively simple environments such as Cartpole and Lunar Lander. It would better to include a broader range of environments, particularly more challenging and high-dimensional ones, to demonstrate Gymnasium's true potential.”
>
> It is worth keeping in mind that performance is not the main focus of Gymnasium - the focus is a shared API. For any given environment, and any given hardware, the performance tradeoff between sync and async vectorization will be a bit different, and in many cases, there will be some alternative, environment-specific approach that works much better. The contribution of Gymnasium is that all these variations can be used under the same API, and easily compared to one another.

---

> > ### Comment · Reviewer_uUKD · 2024-12-03
> >
> > Dear Authors,
> >
> > Thanks for the detailed response! After careful discussions with with other reviewers, I decide to retain my score.

---

### Official Review · Reviewer_nTjF · 2024-11-01

**Soundness:** 4
**Presentation:** 3
**Contribution:** 2
**Rating:** 6
**Confidence:** 2

**Summary:**

The paper provides an overview of the widely-used Gymanasium open-source API for RL experiments. It describes the two approaches for interafcing to environments (standard Env class, and the FuncEnv functional API), discusses the difference between Termination and Truncation, the data types used for action and observation spaces, the role of VectorEnvs in enabling parallelisation of learning. It also reviews the built-in environments, related RL implementations and tool libraries, and alternative API libraries.

**Strengths:**

The paper provides a well-written, comprehensive overview of Gymnasium, which would be of use to a newcomer to this API.

Gymnasium itself is an extremely valuable contribution to the RL community, as evidenced by its widespread adoption.

**Weaknesses:**

There is little original content in this paper as it is basically replicating the documentation available through the Gymnasium website (https://gymnasium.farama.org/index.html).

The decision of whether to accept this paper or not really comes down to a philosophical question around the purpose of ICLR.

The Reviewer's Instructions ask us to consider the paper's value from the contribution of the ICLR community ("What is the significance of the work? Does it contribute new knowledge and sufficient value to the community? Note, this does not necessarily require state-of-the-art results. Submissions bring value to the ICLR community when they convincingly demonstrate new, relevant, impactful knowledge". From this perspective I don't think the paper has much value.  The vast majority of RL researchers that would attend ICLR would already be familiar with, and quite likely already using, Gymnasium. So there is little here that would be especially novel for attendees.

On the other hand we all know that publications at key venues like ICLR have considerable weight for career progression, and undoubtedly the authors deserve credit for their work in developing and maintaining this valuable resource. So there could be a case for acceptance on that basis.

This feels to me like a decision that perhaps needs to be made at a higher level in the conference hierarchy than reviewers?

My personal feeling is that this is perhaps the wrong venue for this work. If it is not accepted, I'd suggest the authors consider submitting to the Software Track of JMLR, as that is designed to support precisely this type of publication.

**Questions:**

I don't have any questions for the authors.

---

> ### Author Response · Authors · 2024-11-21
>
> We would like to thank the reviewer for their valuable comments, and for their very kind words about our work.
> We agree that there is a key “philosophical” question on “What kind of work should be published at ICLR?” that has to be answered somewhere on the organizational level.
> It is worth noting that in the official call for papers, there is a “non-exhaustive list of relevant topics”, which includes:
> datasets and benchmarks
> infrastructure, software libraries, hardware, etc.
>
> For this reason, we believe that Gymnasium fits well within the bounds officially stated for the conference. If the reviewer agrees, we would appreciate an increase in the review rating.
>
> “The vast majority of RL researchers that would attend ICLR would already be familiar with, and quite likely already using, Gymnasium. So there is little here that would be especially novel for attendees.”
>
> It is likely true that it would not be entirely new for attendees. At the same time, things like environment APIs tend to get easily overlooked and forgotten, as the quiet fabric that supports our research. It is often valuable to have a closer look, gain bigger insight into the elements we take for granted, and we believe attendees would benefit from learning more about Gymnasium, even if they already use it in their own work.

---

### Official Review · Reviewer_89ZK · 2024-11-01

**Soundness:** 4
**Presentation:** 4
**Contribution:** 4
**Rating:** 10
**Confidence:** 5

**Summary:**

The paper  introduces Gymnasium, a standardized API designed to improve interoperability and reproducibility in reinforcement learning (RL) research. Gymnasium, an update and extension of the OpenAI Gym, addresses issues such as inconsistent environment implementations and the need for easy experimentation and robust testing. Its main contributions include a unified API compatible with multiple RL libraries, support for both individual and vectorized environments, and novel features such as a functional API closely aligned with Partially Observable Markov Decision Processes (POMDPs), a structured approach to termination and truncation, and expanded support for algebraic spaces. Gymnasium also provides a suite of customizable benchmark environments that range from simple tabular MDPs to complex physical simulations using MuJoCo, helping to accelerate RL algorithm development and testing.

**Strengths:**

- By providing a consistent API, Gymnasium streamlines RL research, making it easier for researchers to compare and build upon previous work.
-The functional API enhances compatibility with theoretical frameworks like POMDPs and enables hardware-accelerated environments using libraries like JAX, which is beneficial for large-scale or computationally intensive applications.
- The built-in vectorization features (Sync and Async) support efficient parallelization of environments, which can significantly improve the performance of RL training processes.
- Gymnasium includes environments from basic MDPs to complex MuJoCo simulations, catering to various RL research needs and supporting a broad range of algorithms and approaches.

**Weaknesses:**

The choice between Sync and Async vectorization modes shows substantial performance variability depending on hardware, which could lead to inconsistencies across different systems or add complexity for users lacking high-performance resources.

**Questions:**

- Does Gymnasium include any tools or guidelines to help users choose the best vectorization mode (Sync vs. Async) based on their hardware?
- How does Gymnasium handle massively parallel or distributed training environments? Is there integration with cloud-based or cluster-based frameworks for larger-scale experiments?

---

> ### Author Response · Authors · 2024-11-21
>
> We would like to thank the reviewer for their valuable comments, and for their very kind words about our work.
>
> “Does Gymnasium include any tools or guidelines to help users choose the best vectorization mode (Sync vs. Async) based on their hardware?”
>
> While there is not an explicit tool, an important feature of the shared API is that from an “external” point of view, sync and async environments behave exactly the same. This way, it is simple to swap “sync” and “async” in a workflow to see the impact of this choice in any researcher’s specific use case.
>
> “How does Gymnasium handle massively parallel or distributed training environments? Is there integration with cloud-based or cluster-based frameworks for larger-scale experiments?”
>
> Gymnasium is entirely agnostic with respect to the execution environment. While there is not a built-in utility for e.g. multi-node vectorization, it is possible to implement such a solution and wrap it in a standard Gymnasium interface, which is then used as usual.

---

### Official Review · Reviewer_WPZ9 · 2024-11-03

**Soundness:** 2
**Presentation:** 4
**Contribution:** 2
**Rating:** 5
**Confidence:** 3

**Summary:**

The paper attempts to address the lack of standardization in reinforcement learning environments. The paper’s library is able to be used in conjunction with a number of libraries, showing that it has been adopted across the field. The paper provides abstraction for environments through a set of fundamental spaces. The paper implements vectorization to enable parallelization across environments. The paper analyzes the increase in environment throughput with vectorization to show that this can provide substantial speedups to using the library.

**Strengths:**

Gym and now gymnasium is an essential part of reinforcement learning with many packages and works being built on top of or inspired by a similar methodology. The abstraction provided by the method is commonly used across reinforcement learning and is easy to use. The vectorization increases throughput of running an environment without significant overhead. The writing is fairly straightforward and it is easy to understand the methodology of the work. Without outside comparison, the results show that the vectorization is able to provide significant speedups. The use of vectorization is key to enable speed-up in performance.

**Weaknesses:**

Despite the use of the name recognition of the library, it is unclear if the improvements from OpenAI Gym to Gymnasium are comparable to the contributions from other RL libraries 2015-present. My main issue with the work is the lack of discussion and comparison with similar work. (In section 2 the paper details a number of comparable works that can use their framework but this is not sufficient to my point.) The paper should complement technical report details with meaningful comparisons to help researchers choose which library to use.

Reinforcement learning is going a massive change in environments due to parallelization but these benefits are from Jax and other GPU engines such as Madrona engine [1], not work created by this library so I do not think it is a fair claim that they can claim credit for those works.

Other works have better vectorization such as Puffer [2] and I would expect a mention or comparison to similar libraries. The paper should explain why one should use Gymnasium versus competitors. It would be helpful to see comparisons across the same environments for each library.

RLlib [3], which is in the related work of the paper, shows substantial deployment and scalability for reinforcement learning on distributed training.

Each of these papers regarding reinforcement learning libraries provide substantial merits for scaling reinforcement learning training. Though I think it is important to encourage RL infrastructure and environment development in reinforcement learning, it is unclear that a set of abstractions provided by Gymnasium have the same level of contribution as the other papers empirical efforts.



[1] Shacklett, B., Rosenzweig, L. G., Xie, Z., Sarkar, B., Szot, A., Wijmans, E., ... & Fatahalian, K. (2023). An extensible, data-oriented architecture for high-performance, many-world simulation. ACM Transactions on Graphics (TOG), 42(4), 1-13.
[2] Suarez, J. (2024). PufferLib: Making Reinforcement Learning Libraries and Environments Play Nice. arXiv preprint arXiv:2406.12905.
[3] Liang, E., Liaw, R., Nishihara, R., Moritz, P., Fox, R., Goldberg, K., ... & Stoica, I. (2018, July). RLlib: Abstractions for distributed reinforcement learning. In International conference on machine learning (pp. 3053-3062). PMLR.

**Questions:**

There is still significant space available for additional results on other environments (which may be more complicated and thus show interesting ablations for synchronous and nonsynchronous methods) provided in Gymnasium. Why not provide additional examples of the work in the paper? There is plenty of room, and it will help show readers unfamiliar with your package about the library's use cases. See weaknesses for examples of papers with more extensive write-ups regarding libraries.

> We hope that this work removes barriers from DRL research and accelerates the development of safe, socially beneficial artificial intelligence.
How does this work do that? This claim is not defended by the experiments.

---

> ### Author Response · Authors · 2024-11-21
>
> We would like to thank the reviewer for their valuable comments, and appreciate the effort they put into the review.
>
> “My main issue with the work is the lack of discussion and comparison with similar work”
>
> It is important to distinguish between the many different scopes that a “reinforcement learning library” could have. There are projects like the mentioned Madrona, which seems to focus on concrete environment implementations. On the other end of the spectrum, there is RLlib, which provides algorithm implementations (which, worth noting, are compatible with Gymnasium). Pufferlib seems to tread the line between environments and interfaces, and Gymnasium almost exclusively focuses on interfaces and utilities.
>
> In the scope of reusable RL environment interfaces, alternatives are rather limited. Some libraries use an ad-hoc abstraction, and some introduce more complete APIs - these are described in Section 2.2. Is there any specific information that the reviewer thinks should be added to Section 2.2?
>
> “Reinforcement learning is going a massive change in environments due to parallelization [...] I do not think it is a fair claim that they can claim credit for those works”
>
> This is a fair statement, but it is important to keep in mind what is the main purpose and scope of Gymnasium, and that is a unified API that can be shared between various projects - both environment and algorithm implementations.
> Without a doubt, there are many environment implementations that will outperform equivalent Gymnasium implementations. But they still can (and often do) expose a Gymnasium interface. This, in turn, makes it possible to instantly use them with Gymnasium-compatible algorithm implementations.
> Consider the counterfactual, i.e. this shared standard does not exist, and a new environment framework gets created, with great performance gains. In order to evaluate the performance of existing algorithms on these environments, it becomes necessary to rewrite the algorithm implementations, whether from scratch or by modifying an existing one.
> Thanks to the shared API provided by Gymnasium, one can instead implement the Gymnasium interface for the environment, and instantly get access to benchmark implementations from e.g. Stable Baselines 3 or CleanRL.
>
>
> “Why not provide additional examples of the work in the paper?”
>
> The main message of the experiment described in the paper is that the vectorization method can be impactful in nontrivial ways. Researchers should be aware of it, and Gymnasium makes it possible to try various approaches and use them under a shared API. We would be happy to include any other experiments that make sense in the context of this paper, but performance scaling is not a central feature of this work.
>
>
> “We hope that this work removes barriers from DRL research and accelerates the development of safe, socially beneficial artificial intelligence. How does this work do that? This claim is not defended by the experiments.”
>
> The key goal of Gymnasium is simplifying and robustifying reinforcement learning research. It does so by freeing up researchers’ time from worrying about environment interfaces and API details, allowing them to focus on their research. This, in consequence, allows research to move forward faster, contributing to the progress of AI development. It is our hope that this progress helps humanity at large.

---

### Meta-Review · Area_Chair_2wiF · 2024-12-16

**Metareview:**

This paper is a technical report of the well-known and widely-used Gymnasium benchmark for Reinforcement Learning (RL). Reviewers have unanimously praised the quality and relevance of the benchmark; however, some expressed concerns about the significance of this report and its overall quality.

Strengths
-----------
- Gymnasium is one of the most important benchmarks for RL;
- It is important that such benchmark is properly described in a technical report.

Weaknesses
--------------
- The report mostly contains technical details that are easily understandable from the online documentation, without providing additional relevant details;
- An extensive and comprehensive benchmarking of different algorithms in the Gymnasium environments is missing.
- The scope of this paper at this venue is questionable.

The paper has received positive scores, with all the Reviewers recognizing the importance of Gymnasium. However, some concerns about the impact and significance of this paper have been raised in the reviews. After engaging into discussion among them and with the meta-Reviewer, Reviewers have agreed that this work does not sufficiently suit this venue. More in detail, Reviewers and meta-Reviewer have agreed that this report does not provide a sufficient number of additional information to provide enough contribution to justify a publication. Moreover, the potential interest of attendees in this paper is likely to be significantly limited, given how well-known the library already is. Finally, the overall quality of the paper is not high, considering that the report mostly describes high-level design choices and technical details, lacking an extensive benchmarking and potential insightful additions, e.g., tutorials.

Based on this, I deem this paper not suitable for publication at this venue. I recommend the Authors considering other venues/journals highly focused on open-source software, e.g., JMLR MLOSS, NeurIPS Call for Dataset and Benchmarks.

**Additional Comments On Reviewer Discussion:**

The points raised by the Reviewer and meta-Reviewer about the significance of this work for this venue have been crucial to come up with a decision about this submission. The final opinion is that other venues/journals focused on publication of technical reports (e.g., JMLR MLOSS, NeurIPS Call for Dataset and Benchmarks) are more appropriate for this paper.

---

### Decision · Program_Chairs · 2025-01-22

Reject